# Trajectory ensembling for fine tuning
# - performance gains without modifying training

**Louise Anderson-Conway** *
Google Research
louiseanderson@google.com

**Vighnesh Birodkar**
Google Research

**Saurabh Singh**
Google Research

**Hossein Mobahi**
Google Research

**Alexander A. Alemi**
Google Research

## Abstract

In this work, we present a simple algorithm for ensembling checkpoints from a single training trajectory (trajectory ensembling) resulting in significant gains for several fine tuning tasks. We compare against classical ensembles and perform ablation studies showing that the important checkpoints are not necessarily the best performing models in terms of accuracy. Rather, relatively poor models with low loss are vital for the observed performance gains. We also investigate various mixtures of checkpoints from several independent training trajectories, making the surprising observation that this only leads to marginal gains in this setup. We study how calibrating constituent models with a simple temperature scaling impacts results, and find that the most important region of training is still that of the lowest loss in spite of potential poor accuracy.

## 1 Introduction

Ensembling is by now well-established as a method to improve performance as well as robustness of a model [6, 16] (for a recent review, see Ganaie et al. [7]). A drawback is the increased computational cost that comes from independently training several models. Ideally, one would like to obtain similar performance gains to ensembling independent models at the computational cost of training only one. In this work, we make progress towards this goal by forming ensembles from only one training run for fine tuning, showing significant gains.

We study ensembling for the problem of image classification [21, 1]. Existing methods generally require costly modifications [11, 8, 23, 14, 17, 24] to training. In the context of semi-supervised learning, performance gains have been seen using temporal ensembles and mean teachers, which are computationally efficient methods, however, no gain is observed in the fully supervised setting using this approach [15, 22].

In this work, we present an *algorithm that results in performance gains without modifying training in the setting of fine tuning*. We show that increased accuracy can be obtained by ensembling checkpoints from one training trajectory, and present an algorithm for selecting checkpoints to ensemble. This is reminiscent of the results obtained in the context of training-from-scratch in checkpoint ensembles [4, 12, 20] though we extend this work to the realm of fine tuning, as well as ensembles of mixtures of checkpoints gathered from many different runs and at various times in training, which we dub "combination ensembles".

---

*Work done as part of the Google AI residency program

Has it Trained Yet? Workshop at the Conference on Neural Information Processing Systems (NeurIPS 2022).

Traditional fine tuning is a careful balance between retaining the knowledge of the pretrained model vs creating a model that is well-trained on the distribution of interest. By creating an ensemble instead, we can combine the best of these worlds, and see significant ($\gtrsim 1\%$) gains over existing benchmarks.[2] Furthermore, we are able to gain a better understanding of the underlying principles of the gains coming from ensembling. It turns out, rather than simply choosing the most accurate models, relatively poor models with low loss are vital for forming the best-performing ensembles, which we will discuss more throughout this work.

## 2   Approaches and experiments

For our experiments, we fine-tune a ResNet50 [10], pretrained on imagenet [19], on tasks that have been outlined in the (lightweight) Visual Task Adaptation Benchmark (VTAB) [27]. We consider three data sets (sun397 [25], dtd [5] and diabetic retinopathy detection [13]), and save checkpoints from every epoch from multiple runs. In the following sections, we explore mixtures of ensembles of such checkpoints, and present an algorithm for a simple technique that yields significant improvement in accuracy over the VTAB benchmark.

---

**Algorithm 1** − forming an $M$-member ensemble from epochs $N_1...N_M$

---

1: **Train** on **train split**, saving every $n$ epochs ($n$ is often taken to unity but can be larger to speed up the process).
2: Select the *checkpoint that performs the best on the validation set*.
3: Pick $M-1$ checkpoints by random sampling from the region around the **lowest validation loss**.
4: Create a set of ensembles, (by resampling around the region of lowest validation loss), **evaluate** on the **validation set**, saving the best-performing $N_i^{best}$.
5: **Retrain** on **train + val**, saving checkpoints from the same epochs as the best-performing ensemble on the validation set..
6: Evaluate this ensemble on the test set.

---

Using this algorithm, we will show gains over the VTAB lightweight benchmarks of around or above one percentage point, presented in section 3. We define our ensemble output as follows:

$$\mathcal{E} = \frac{1}{M} \sum_{i=1}^{M} w_i \, \text{softmax}\,(y_i) \ , \qquad \text{softmax}(y_i) = \frac{e^{y_i}}{\sum_j e^{y_j}} \tag{1}$$

where $y_i$ denotes the output of the model after being finetuned for $N_i$ epochs. The ensembling is therefore here done entirely in *probability space*. The $w_i$ are scalar weights, normalised such that $\sum_{i=1}^{M} w_i = M$. For our main results, we consider the simplest case of unit weights [3].

We will present an experiment where we do learn optimal weights in section 3.1 as an ablation study, giving more support to the region described in point 3 in the algorithm above indeed being the vital region for ensembling. For further details, see appendix C.

## 3   Results: Trajectory ensembling leads to significant ($> 1\%$) gains

Using algorithm 1 on the tasks outlined in the VTAB lightweight paper, we *see a gain on all three considered data sets over the corresponding benchmarks, often in excess of* $1\%$, as illustrated in table 1. The ensemble performance given in table 1 is that of the five-member ensemble that performs best on the validation set out of five sample ensembles, then evaluated on the test set, as described by the algorithm in section 2. We also consider other number of ensemble members in appendix C, which also contain technical details of the ensembling procedure. For details on how training was done, see appendix B.

Analysis were done with both cosine- as well as step-wise constant learning rate schemes. We find that cosine decay produces the best results for sun397 and retinopathy, and step-wise decay for

---

[2]For complementary approaches to ensembles in the context of fine tuning, see [18] and works therein.
[3]We have also considered training for "optimal weights" on a held-out validation set using cross-entropy loss, but have seen no significant performance increase from this to merit the computational cost.

Table 1: Main results

| data set | sun397 | dtd | retinopathy |
|---|---|---|---|
| VTAB | 70.7 | 74.0 | 79.3 |
| VTAB + data aug. | 72.6 (0.05) | 74.8 (0.1) | 82.3 (0.1) |
| 5-member ensemble | **73.5** (0.2) | **75.7** (0.2) | **84.0** (0.1) |

`dtd`. For other schemes and numbers of ensemble members, see appendix C. All numbers are means of 5 independent runs and numbers within parenthesis denote one standard deviation.

Note that the best result of a VTAB sweep might not always come from the same run as the best-performing ensemble.

## 3.1 Ablation studies: the best ensembles use early checkpoints, even if calibrated

As noted in section 2, the checkpoints to include in these ensembles comes from relatively early in the training. In this section, we provide ablation experiments to justify this. This is done by a stacking approach to ensembles [28, 26] where we assume oracle-level access to the test set on `sun397`. We form an ensemble including a large number ($\simeq 30$) of checkpoints throughout training, and learn optimal weights for these checkpoints by minimising the cross-entropy loss with a regularisation term, (equation (3)), (for details, and qualitatively similar results on `dtd`, see appendix C.2).

We find a peak in these optimal weights *early in training*, coinciding with the region of lowest loss − the precise region of interest in our algorithm 2, see figures 1, 2. Note that the ensemble members obtaining the highest learned weight *does not* correspond to those with the highest accuracy, and that these learned, large, ensembles only marginally outperforms our 5-trajectory ensemble at 73.9% accuracy, but is outperformed by trajectory ensembles of $\simeq 10$ members (see table 4).

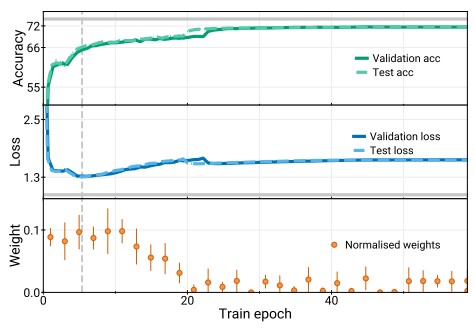
(a) Learned ensemble weighs.

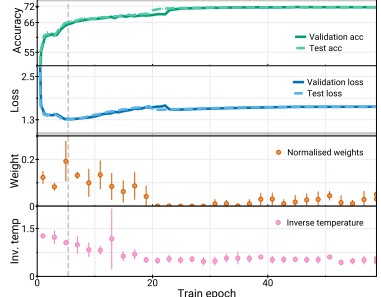
(b) Learned weights and inverse temperatures.

Figure 1: A clear peak in the weights (orange circles ) can be seen around the region with the lowest test- and validation loss (blue curves), though this does not correspond to the highest accuracy (green curves). This holds true even with temperature scaling (right subfigure, where inverse temperature correspond to pink circles). Grey solid horizontal lines denote learned ensemble accuracy and loss respectively. We see marginal performance gains from including temperature scaling (from 73.9% to 74.3%), and dashed grey vertical line denotes the point of lowest test loss. All measures are a mean of five samples and error bars denote one standard deviation.

This serves as justification to our inclusion of the best-performing checkpoint in the ensemble if one wishes to maximise gains in accuracy in section 2, though this is an ad-hoc procedure since accuracy is non-differentiable and we cannot perform a version of SGD to maximise for it directly.

One reason for this discrepancy between low loss and high accuracy could be that the models are poorly calibrated. We investigate this by introducing a temperature scaling [9], simultaneously learning model calibration and ensemble weights. This amounts to introducing an additional parameter

for each checkpoint, $\beta_i$, and modifying the standard cross-entropy loss function (3) to:

$$\mathcal{L} = -\sum_{x \in \chi} \text{label}(x) \log \left( \frac{\sum_i |w_i| \sigma \left( \beta_i \text{logit}_i(x) \right)}{\sum_j |w_j|} \right) + r \sum_i |w_i|, \qquad (2)$$

where the regularisation coefficient $r$ is taken to $10^{-4}$, though experiments do not seem very sensitive to this exact number (experiments of $r \in [10^{-2}, 10^{-5}]$ have been done without qualitative difference). We find that this reduces the early peak, but does not remove it, suggesting this is not a satisfactory explanation for the observed phenomena (see figure 1). We see marginal performance gains from including temperature scaling (from $73.9\%$ to $74.3\%$).

### 3.2 Comparison to traditional- and combination ensembles

In this section, we compare our results to classical ensembles, where one combine the final/best checkpoints from several independent runs, as well as combination ensembles, where one combines multiple checkpoints from multiple independent runs.

We find that our simple trajectory ensemble of section 3, formed according to algorithm 1, *outperforms classical ensembles*. However, learning ensembles across many runs do result in marginal performance gains compared to a learned trajectory ensemble. Results of these investigations are displayed in table 2. Note here that these are best checkpoints from within the same run as forming the ensemble, not the best result of the VTAB sweep, as presented in table 1.

Table 2: Comparison of various ensembling techniques on the `dtd` data set.

| **Method:** | best ckpt | 5-member trajectory ensemble | traditional ensemble | learned trajectory ensemble | learned combination ensemble |
|---|---|---|---|---|---|
| # ckpts / run | 1 | 5 | 1 | 37 | 37 |
| # runs | 1 | 1 | 5 | 1 | 5 |
| test acc. (%) | 73.9 | 75.7 | 74.6 | 76.3 | 77.6 |

## 4 Analysis, future directions and limitations

In this work, we present a *simple algorithm for forming trajectory ensembles that leads to a gain in accuracy of $\gtrsim 1\%$* using trajectory ensembling for several fine-tuning tasks that exhibit a global minima in the validation loss at an early times. We carry out ablation studies justifying our algorithm and compare results to classical ensembles as well as ensembles combining various checkpoints from several independent runs (combination ensembles).

We show that this early region of lowest loss is vital for our such trajectory ensembles, and we point out that this region *does not* always correspond to high accuracy, as is the case in the `sun397` data set. Including a simple temperature scaling to calibrate models does not resolve this discrepancy.

As a result of this discrepancy between loss and accuracy, the learned ensembles in our ablation experiments are sometimes outperformed by our simple trajectory ensembles formed using algorithm 1. This shows it is also vital to include the best-performing checkpoints in the ensemble.

Including checkpoints from several different runs seem to lead to only marginal gains at a significantly increased computational cost. A limitation of this work is that all fine tuning is done starting from the same pretrained model, and it would be interesting to consider ensembles of various pretrained models, combining the approach taken herein with ideas from Mustafa et al. [18]. Another promising direction for further work would be to consider how these techniques generalise to a larger variety of data sets and architectures.

An important upside of this technique is that it reduces the sensitivity of performance on hyperparameters such as learning rate scheme and number of steps trained, which otherwise are vital for fine tuning performance.

Given the remarkable simplicity of forming efficient trajectory ensembles in combination with their demonstrated performance gains, we encourage others to try trajectory ensembles.

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

# A  Experimental details - pretraining

All experiments in this work is done using ResNet50 [10] on ImageNet [19] using the big_vision framework [2]. This training is a standard pretraining, and results in a model with accuracy 76.3% which is similar to standard models of this kind, (for ex the Keras ResNet50v2 model reaches an accuracy of 76.0%).

# B  Experimental details - finetuning

Fine tuning is done following the Visual Task Adaptation Benchmark lightweight training setup [27]. It is done on an $4 \times 4$ Jellyfish TPU with a batch size of 512 for all data sets. This is trained using SGD with momentum (0.9), all implemented in JAX [3]. We use moderate data augmentation consisting of left-right flips and random crops of a $224 \times 224$ region of images of $256 \times 256$ for step-wise decaying learning rate scheme (as is done in the original VTAB paper), and of $256 \times x$, where $x \geq 256$ and chosen to perform proportions of the image for cosine learning rate scheme, which is standard in the updated big_vision code for fine tuning [2].

The lightweight VTAB sweep consists of a sweep over four different hyperparameter combinations: two different initial learning rates and two different number of training steps. We sweep over initial learning rates $[0.1, 0.01]$ and total number of training steps $[2, 500, 10, 000]$, and use a three-way split of the data set: train, validation and test. The model is first trained with these four combinations of hyperparameters on the train set and evaluated on the validation set. The best-performing combination of hyperparameters is then run again, now training on train + validation, and the final accuracy is then reported on the test set. The step-wise learning rate scheme decreases the learning rate by a factor of 10 after $[1/3, 2/3, 9/10]$ of training. Both learning rate schemes used here have a linear warm-up of 500 steps.

This is the approach we follow this methodology for our ensembles as well: Separate training runs are done on only the train set, as well as the train+validation data set for use with ensembling. From each of these training runs, we save checkpoints each epoch, as well as the checkpoint from the last train step.

# C  Experimental details - ensembling

## C.1  Simple trajectory ensembles

For our simple trajectory ensembles formed using the algorithm in section 2, no active training is required. The region around the lowest validation loss described in 2 is chosen to be such that we sample from epochs between $E_{\text{initial}}$ and $E_{\text{final}}$, where $E_{\text{initial}}$ is defined to be the first epoch during training where validation loss gets $\delta L_{\text{initial epoch}}$ away from the minimal loss, that is, the first epoch with validation loss $L_{\text{initial epoch}} < L_{\text{min}} + \delta L_{\text{initial epoch}}$. Similarly, the final epoch in the sampling region is defined as the last epoch during training where $L_{\text{final epoch}} < L_{\text{min}} + \delta L_{\text{final epoch}}$.

Dpresented in section 3 is acquired by selecting such a region to sample from around the lowest loss, specific values for $\delta L_{\text{initial epoch}}, \delta L_{\text{final epoch}}$ are presented in the table below 3. In all cases, we use a batch size of 16 at evaluation.

It is often convenient to choose $L_{\text{initial epoch}} \lesssim L_{\text{final epoch}}$. This is due to the fact that these choices are done for a round where the model was trained on just the train set, whereas the final model will be trained on train + validation. This means, the final model will be trained on more samples, and the lowest validation loss is likely to happen slightly later in training, which means we would like to cover more of the region after the lowest validation loss has been reached than before.

## C.2  Learned ensembles

For learned trajectory ensembles, as illustrated in section 3.1, we use roughly 30 models (for `dtd` 37, for `sun397` 31, models equidistantly sampled throughout training and we form an ensemble of all of these models. In this, we assume oracle-level access to the test set, and form a trajectory ensemble from a run that has been fine-tuned on both training + validation data. We use $\min(2000, \ 0.1 \cdot \texttt{num}(\text{test samples}))$ to learn the weights for this ensemble.

Table 3: Values of loss from which ensemble members were sampled from.

| data set | sun397 | dtd | diabetic retinopathy |
|---|---|---|---|
| $\delta L_{\text{initial epoch}}$ | 0.15 | 0.6 | 0.1 |
| $\delta L_{\text{final epoch}}$ | 0.1 | 0.2 | 0.1 |

The weights here are parameterised to emulate the situation in 2, and are constrained to be positive and normalised, and the sum of absolute values are added for regularisation purposes. The optimization does not seem to be very sensitive to the value of the regularisation coefficient $r \geq 0$ below, and it is taken to be small.

The loss function used is:

$$\mathcal{L} = -\sum_{x \in \chi} \text{label}(x) \log \left( \frac{\sum_i |w_i| p_i(x)}{\sum_j |w_j|} \right) + r \sum_i |w_i|, \tag{3}$$

and the optimization is done using Adam with a learning rate of $10^{-4}$, and training for 1000 train steps on `sun397`, 400 steps on `dtd`. Sweeps of other choices of learning rates/train steps were done as well, and a combination that allowed for training to stabilise was chosen. A sweep of the regularisation coefficient in equation (3) is done with $r \in [10^{-3}, 10^{-4}, 10^{-5}]$, though no qualitative difference is found between resulting weights. In the main text, all experiments presented were done with a regularisation of $10^{-4}$.

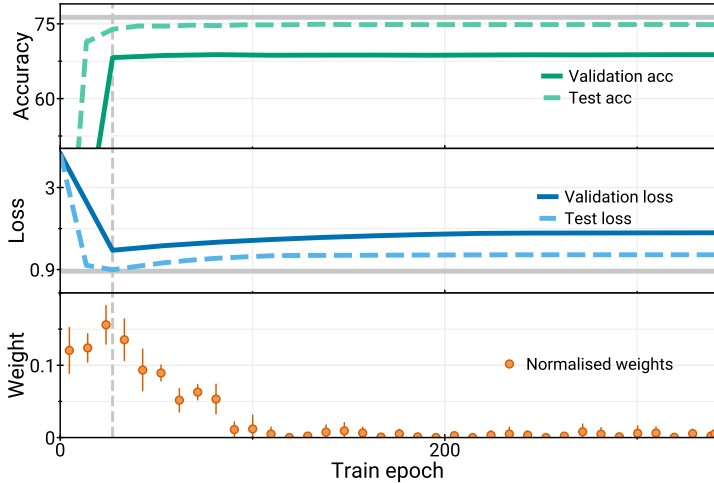

Figure 2: Results for fitting optimal weights to the test set of the descriptive textures data set (`dtd`). As in figure 1, we see a clear peak in weights (lowest figure) corresponding to the area with the lowest loss (middle blue curves), once more coinciding with our sample region centered around the lowest loss (vertical grey dashed line) for our simple algorithm Grey horizontal lines denote the accuracy and loss of the learned ensemble.

Results for `dtd` are not qualitiatively different from those on `sun397` (presented in the main text, figure 1). Rather, in the `dtd`-case (figure 2), the curves for loss and accuracy are much less complicated than those for `sun397` (figure 1), and maximal accuracy is reached quite early on and thereafter remains constant. In order to maximise model difference, and thus ensemble gains, we therefore choose the checkpoint with the highest accuracy *latest in training*, again supporting our algorithm in section 2.

### C.2.1 Learning ensembles with temperature scaling

For training weights and inverse temperatures simultaneously, the optimization is more difficult, and we adjust learning rate/number of steps to accommodate this, now using a learning rate of 0.01 and training for 400 steps. All results presented are means of five independent runs.

### C.2.2 Learned ensembles from multiple runs

When we learn ensembles coming from multiple runs (table 2), we once more find that the majority of the weights remain on early, low-loss checkpoints, rather than final checkpoints for each model, see figure 3 for an illustration [4].

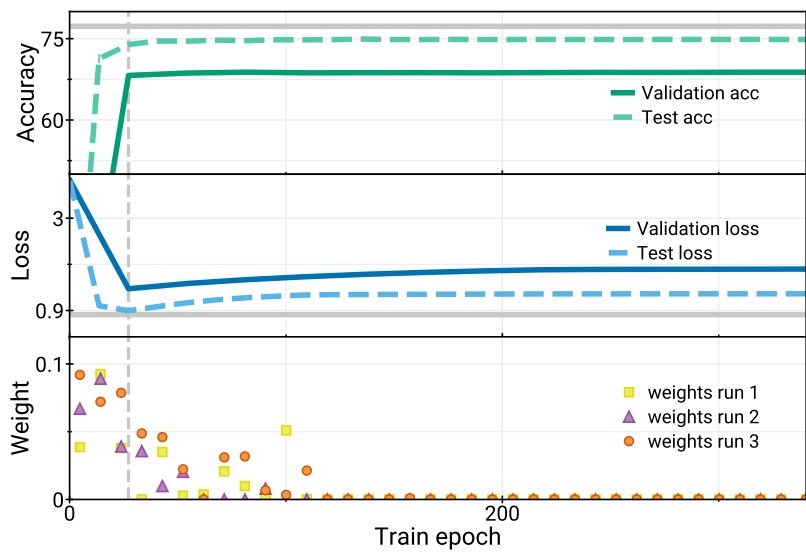

Figure 3: Fitting weights of a combination ensemble from three different runs still results in peaks at early times (lower scatter plots), and close to zero weights assigned to later models.

### C.3 Diminishing return on additional ensemble members

In addition to the 5-member ensembles presented in the main text (table 1), we carry out experiments with other number of ensemble members for both learning rate schedules (tab 4, 5). In both cases, we can clearly see a diminished returns on adding in additional ensemble members, which agrees with results of [24]. This furthermore illustrate the powerful benefit trajectory ensembles can have to desensitise a run to choice of hyperparameters. As seen in the case of `dtd` above, even though the run starts of almost a full $5\%$ under the best-performing checkpoint (here coming from another choice of hyperparameters in the VTAB lightweight sweep setup), it only takes a 3-member trajectory ensemble for the trajectory ensemble to beat the best checkpoint from the entire hyperparameter sweep.

---

[4]This is done for checkpoints from 3 independent runs, though no qualitative difference is seen when including more independent runs in the mixture.

Table 4: Ensemble top-1 accuracy for trajectory ensembles with various number of members when trained with cosine learning rate scheme. All values are a mean of five runs and numbers within brackets denote one standard deviation.

|  | sun397 | dtd | retinopathy |
| --- | --- | --- | --- |
| final ckpt of run | 72.22 (0.15) | 71.29 (0.73) | 81.01 (0.04) |
| best result from VTAB sweep | 72.62 (0.05) | 75.20 (0.24) | 82.31 (0.11) |
| 2-member trajectory ensemble | 72.96 (0.29) | 73.31 (0.87) | 82.96 (0.20) |
| 3-member trajectory ensemble | 73.23 (0.25) | 75.71 (0.73) | 82.95 (0.25) |
| 4-member trajectory ensemble | 73.45 (0.20) | 74.07 (0.75) | 83.57 (0.12) |
| 5-member trajectory ensemble | 73.52 (0.15) | 74.55 (0.58) | 83.99 (0.10) |
| 6-member trajectory ensemble | 73.78 (0.09) | 75.68 (0.30) | 83.75 (0.07) |
| 7-member trajectory ensemble | 73.87 (0.19) | 75.66 (0.16) | 83.91 (0.06) |
| 8-member trajectory ensemble | 73.87 (0.17) | 76.31 (0.37) | 83.92 (0.04) |
| 9-member trajectory ensemble | 73.99 (0.15) | 75.99 (0.56) | 84.00 (0.06) |
| 10-member trajectory ensemble | 74.07 (0.17) | 75.52 (0.42) | 84.01 (0.07) |

Table 5: Ensemble top-1 accuracy for trajectory ensembles with various number of members when trained with step-wise decaying learning rate scheme. All values are a mean of five runs and numbers within brackets denote one standard deviation (for 10-member ensmeble on sun397, there were not enough checkpoint in the span defined by 3 so we had to marginally expand the sample region).

|  | sun397 | dtd | retinopathy |
| --- | --- | --- | --- |
| final ckpt of run | 71.65 (0.12) | 70.36 (0.38) | 80.01 (0.30) |
| best result from VTAB sweep | 72.35 (0.05) | 74.84 (0.14) | 80.40 (0.05) |
| 2-member trajectory ensemble | 72.24 (0.22) | 74.57 (0.33) | 81.23 (0.12) |
| 3-member trajectory ensemble | 72.51 (0.21) | 74.09 (0.40) | 81.19 (0.13) |
| 4-member trajectory ensemble | 72.90 (0.17) | 75.12 (0.24) | 81.26 (0.12) |
| 5-member trajectory ensemble | 73.12 (0.16) | 75.72 (0.21) | 81.63 (0.09) |
| 6-member trajectory ensemble | 73.21 (0.10) | 75.69 (0.21) | 81.69 (0.14) |
| 7-member trajectory ensemble | 73.34 (0.14) | 76.02 (0.54) | 81.69 (0.12) |
| 8-member trajectory ensemble | 73.33 (0.16) | 75.60 (0.23) | 81.52 (0.09) |
| 9-member trajectory ensemble | 73.45 (0.21) | 76.20 (0.30) | 81.35 (0.08) |
| 10-member trajectory ensemble | 73.59 (0.23) | 75.95 (0.24) | 81.64 (0.09) |

