# OpenReview forum: "Trajectory ensembling for fine tuning -  performance gains without modifying training"
_NeurIPS.cc/2022/Workshop/HITY — HITY Workshop NeurIPS 2022_

### Official Review · Reviewer_tEge · 2022-10-05
**A method to form trajectory ensembles from multiple checkpoints**

**Rating:** 1
**Confidence:** 1

**Review:**

This paper introduces a simple algorithms to form trajectory ensembles from multiple checkpoints. To this, end multiple checkpoints are chosen from which the ensemble is computed. Interestingly, the most important checkpoints are not necessarily the best performing ones in terms of accuracy. Also, the gain from several independent trajectories is limited.
The writing is reasonably clear and fits the workshop. Accept.

---

### Official Review · Reviewer_nQp1 · 2022-10-18
**Issues with methodology and clarity, but presents 2 interesting questions**

**Rating:** 1
**Confidence:** 2

**Review:**

In general, it is difficult to follow the many claims being made throughout the paper, and to what extent they are sustained by the experiments and observations made.

The paper presents the interesting claim that ensembling during the fine-tuning process can help with OOD and catastrophic forgetting, but this claim is  not substantiated.

Another claim is: "We show that increased accuracy can be obtained by ensembling checkpoints from one training trajectory". This has been already shown in the literature, particularly some of the cited sources like Garipov et al., 2018. In any case, the claim that ensembling snapshots with better validation loss is beneficial is already known. Furthermore, there is a mention of inefficiency of traditional ensembling procedures, but the relevance of runtime in this work is unclear (beyond single-path ensembling).

A main claim is that the presented algorithm results in performance gains. To me it is unclear to what extent this algorithm is any better than FGE (Garipov et al., 18): it appears to be slower, since it requires to retrain, and if it is better it may be to reasons like increased training budget, model capacity, or training on validation data. On this note, the evaluation procedure raises concerns: models are being trained on cross-validation data, hinting at the risk that test data may be being over-used, and gains are reported for the best-of-five runs, when normally averages with standard error are reported, so results that are reported as significant lack a clear definition of said significance. A similar concern applies to the results from table 2. In summary, there are many possible explanations for the observed increase in performance other than the proposed algorithm, and the results as presented do not compel to think otherwise.

An interesting claim is: "Rather than simply choosing the most accurate models, relatively poor models with low loss are vital for forming the best-performing ensembles". To substantiate this, the authors learn an ensemble out of 30 models using a combination of training and heuristics. I see several methodological issues: First of all, the ensembling is being performed via a softmax on the top of the logits.  Softmax exponentially saturates the low entries towards zero, and this effect may be much stronger than the linear weights, making them irrelevant. Finally, the results may be specific to this paper's setup. For this reason it can't be generally concluded that "models with low loss are vital", but the results are encouraging and warrant more attention.

I recommend the authors to reduce the number of claims, and to design experiments that specifically target those in a more concise way, removing confounding factors like using different training budgets or potentially overfitting setups. The raised question whether ensembling through fine-tuning can robustify against OOD and catastrophic learning is a very interesting one. More comprehensive and exhaustive experiments on the "poor loss, high weight" hypothesis could also be interesting.

I propose to low-confidence accept because raising these last 2 questions can be useful for the comunity, but I consider that the issues with clarity and methodology are concerning.

---

### Decision · Program_Chairs · 2022-10-20

Accept